

# Characterization of social frailty domains and related adverse health outcomes in the Asia-Pacific: a systematic literature review

Tengku Aizan Hamid[1,*], Sarah Abdulkareem Salih[2,*],
Siti Farra Zillah Abdullah[1], Rahimah Ibrahim[1,3] and Aidalina Mahmud[4]

[1] Malaysian Research Institute on Ageing (MyAgeing™), Universiti Putra Malaysia, Serdang, Selangor, Malaysia
[2] Department of Architecture, Faculty of Design and Architecture, Universiti Putra Malaysia, Serdang, Selangor, Malaysia
[3] Department of Human Development and Family Studies, Universiti Putra Malaysia, Serdang, Selangor, Malaysia
[4] Faculty of Medicine and Health Sciences, Universiti Putra Malaysia, Serdang, Selangor, Malaysia
* These authors contributed equally to this work.

Corresponding author
Sarah Abdulkareem Salih,
sarah_salih@upm.edu.my

## ABSTRACT

**Background:** Frailty is a significant healthcare challenge worldwide, increasing interest in developing more assessment tools covering for frailty. Recently, there has been a growing awareness of a correlation between social variables and frailty in older people. However, there is a lack of understanding of the social domains of frailty and the related adverse outcomes, particularly in the Asia-Pacific settings. This study aimed to characterize the social frailty domains and their health outcomes by overviewing the frailty screening tools in older people living in the Asia-Pacific region.

**Methodology:** A systematic review, using the PRISMA guideline, was conducted on articles published between 2002 and 2023 from three electronic databases: PubMed, Scopus, and ScienceDirect. A manual search was conducted for the references of the included articles using Google Scholar. Included articles must be in English and were based on empirical evidence published in peer-reviewed journals and focus on the assessment of domains of social frailty in older people aged 60 or over in the Asia-Pacific (East Asia, Southeast Asia, and Oceania).

**Result:** A total of 31 studies were included in the thematic analysis, from which 16 screening tools measuring six social domains were reviewed. The six domains were: social networks, followed by social activities, social support, financial difficulties, social roles, and socioeconomic, arranged in four categories: social resources, social needs, social behaviors (or social activities), and general resources. The six social domains predicted mortality, physical difficulties, and disability incidence. Other adverse health outcomes were also associated with these social domains, including cognitive disorders, mental illness, and nutritional disorders ($n = 5$ domains each), dementia ($n = 4$ domains), and oral frailty, hearing loss, obesity, and chronic pain ($n = 3$ domains each).

**Conclusion:** Overall, social frailty is a complex construct with multiple dimensions, including the frailty of social and general resources, social behaviors, and social needs, leading to several health disorders. The findings contribute to understanding

the conceptual framework of social frailty in older people and its related health outcomes. Therefore, it could facilitate professionals and researchers to monitor and reduce the risks of adverse health outcomes related to each domain of social frailty, contributing to a better aging process.

## INTRODUCTION

Global aging and longevity contribute to an increase in the number of people with multidimensional frailty and complex care needs. The issue of frailty in older people has received increasing scientific attention in aging research and elderly clinical care (*Gobbens et al., 2010*). Consequently, the concept of frailty in older adults has become increasingly relevant to geriatrics and gerontology (*Chao & Huang, 2015*). One of the most well-known conceptual models of frailty is the (unidimensional) biological model, which refers to an individual's physical functioning and includes five indicators, namely exhaustion, weight loss, slowness, weakness, and low energy expenditure (*Bessa, Ribeiro & Coelho, 2018*; *Zhang et al., 2023*). Cognitive and psychological frailty is also relatively common in older adults with various health and independence problems (*Tsutsumimoto et al., 2017*). Previous research on frailty in older adults showed that the most commonly identified dimensions are physical and cognitive while lacking clarity on the social extent (*Akasaki et al., 2022*; *Bunt et al., 2017*). Social frailty is the least covered dimension in the existing literature. The physical and cognitive models of frailty and their indexes predict a combination of symptoms, diseases, and conditions with a one-dimensional perspective (*Henry et al., 2023*; *Hironaka et al., 2020*).

Contemporary research from the Asia Pacific region on frailty in older adults assumes a complex interaction between various dimensions, such as physical, cognitive, psychological, environmental, and social domains (*Henry et al., 2023*; *Lee et al., 2022*). Frailty in the multidimensional model refers to a lack of one or more operating dimensions of human resources induced by the effect of a number of variables (*Gobbens et al., 2010*). However, a clear conceptualization model of social frailty has not yet been developed (*Duppen et al., 2019*). Existing studies from the Asia Pacific mentioned that social frailty could describe a lack of frequent participation in social events, networks, and contact and insufficient support, leading to serious health outcomes. Separately, these studies reported that social frailty in community-living older adults could be associated with a low quality of life (*Ong et al., 2022*) and various health issues, including the risk of disability and mortality (*Makizako et al., 2015*), dementia (*Bae et al., 2018*), physical function (*Watanabe et al., 2020*), chronic pain (*Hirase et al., 2019*), and Alzheimer's disease (*Tsutsumimoto et al., 2019*). Furthermore, other dimensions of frailty, such as physical and cognitive, were identified as the main factors associated with social frailty (*Bae et al., 2018*; *Hironaka et al., 2020*; *Teo et al., 2019*; *Tsutsumimoto et al., 2017*). Social frailty and deficits could also refer

to deficiencies in living situations, socioeconomic status, and social roles (*Lee et al., 2020*). In their review of social frailty in older adults, *Bessa, Ribeiro & Coelho (2018)* mentioned that social defect is multidimensional, with a range of social resources and domains that can meet social needs. Thus, social frailty can be defined as the loss of one or more human social resources essential for fulfilling basic human needs throughout life (*Bunt et al., 2017*). Previous studies confirmed that social relationships and support are the most promising resources or dimensions of social frailty (*Lee et al., 2020*; *Faller et al., 2019*). Besides, reduced participation in social and community activities, such as volunteering, visiting family and friends, attending social events, and participating in social community clubs, can lead to social frailty (*Bessa, Ribeiro & Coelho, 2018*; *Pek et al., 2020*; *Teo et al., 2017*). Some studies also described economic insecurity as a critical domain of frailty (*Armstrong et al., 2015*; *Lee et al., 2020*). However, there is a lack of evidence on the exact domains of social frailty from a comprehensive perspective that involves several assessment tools (*Bessa, Ribeiro & Coelho, 2018*; *Lee et al., 2020*).

In contrast, it is worth noting that the Asia Pacific population is ageing faster than any other region in the world, accounting for 60% of the global population in 2022 and an estimated 55% by the year 2050 (*Statista, 2023*). It is projected that the number of older adults in Asia Pacific who are aged 60 and over will rise from 549 million in 2017 to around 1.3 billion by the year 2050 (*Statista, 2023*). This increased number of older adults may lead to the prevalence of frailty status, growing burdens on society, and various health outcomes (*Salih et al., 2023*; *Abe et al., 2019*; *Teo et al., 2017*). However, despite the growing evidence on the relationship between frailty, social deficits, and adverse health outcomes in older adults (*Hirose et al., 2021*), there is a significant knowledge gap regarding the social dimensions of frailty and the possible health outcomes related to each domain, especially in the Asia-Pacific context (*Faller et al., 2019*). Besides, there is a need to provide a better understanding of social frailty domains and their role in screening tools and related adverse outcomes for the Asia-Pacific region, which has the largest population of older adults globally. Therefore, the current study aimed to characterize the social frailty domains and their health outcomes by overviewing the frailty screening tools in older people in Asia-Pacific settings.

This systematic literature review contributed to the existing knowledge on the wide range of social frailty domains of elderly frailty and related adverse health outcomes to each domain by reviewing the frailty screening tools in Asia Pacific settings. Therefore, it contributed to a reference for health professionals and researchers in making informed decisions by linking social health science to public health and health sciences.

## Theoretical framework and conceptual background of the study

The conceptual bases of this study were developed from Gobbens's and Bunt's frameworks on the determinants of social frailty (*Bunt et al., 2017*; *Gobbens et al., 2010*). *Gobbens et al. (2010)* suggested three main frailty domains: physical, psychological, and social. *Gobbens et al. (2010)* highlighted two sub-domains for social frailty, including social network and social support; achieving these domains enhances the quality of life and well-being as people age. *Bunt et al. (2017)* concept of social frailty developed based on the social needs

concept of Social Production Function theory. Based on this theory, *Bunt et al. (2017)* confirmed that social frailty in old age is affected by four social factors: social needs, social resources (such as social supports and social networks), social behaviors or social activities, and general resources (indirect way of fulfilling social needs, such as living situation, educational level, and income or financial status). As a result, the context of this review was applied to the selected studies and assessment tools to extract data on the suggested social domains for social frailty.

## Rationale of the systematic review

Elderly social frailty is an emerging health challenge associated with various health outcomes (*Abe et al., 2019*). Even though many review studies (*Faller et al., 2019*; *Bunt et al., 2017*; *Bessa, Ribeiro & Coelho, 2018*) conducted on older people's frailty, the field still lacks comprehensive systematic literature reviews that integrate social frailty domains and their health outcomes. The existing reviews only assessed the social dimension of frailty and did not characterize the relevant health outcomes to each social domain. To address this gap, the current study systematically reviewed 31 journal articles selected from 1,896 text materials to address this gap. It followed a rigorous systematic method to ensure the future replicability of results by employing a clear review protocol and limiting bias in results by only focusing on empirical evidence (*Moher et al., 2009*). It also systematically analyzed the relevant scientific findings of the selected journal articles and showed whether they are consistent and can be generalized across settings or whether results vary significantly by subsets.

To describe the frailty syndrome in older people, existing review studies focused on reviewing the findings of the frailty assessment tools (*Duppen et al., 2019*; *Bessa, Ribeiro & Coelho, 2018*). Despite the unmanageable amounts of information on the frailty assessment tools, there is a need for systematic reviews to integrate efficiently existing knowledge to understand better the adverse health outcomes related to each social domain and contribute to providing data for rational decision-making by healthcare providers and policymakers. There is a significant knowledge gap in the Asia-Pacific context regarding frailty's social dimensions and the possible health outcomes related to each domain (*Faller et al., 2019*). There is, therefore, a need to provide a better understanding of social frailty and its domains and health outcomes in Asia-Pacific settings, the world's largest population of older people. Thus, the current study focused on the frailty tools that measure social frailty and its health outcomes in Asia-Pacific settings.

More specifically, the current systematic review contributed to (1) being one of the early systematic literature reviews that covered a wide range of social domains and determined the related adverse health outcomes to each domain, (2) reviewing the existing frailty screening tools in the Asia Pacific settings and characterize social domains and relevant adverse outcomes, (3) a better understanding of the existing theoretical model of social frailty proposed by existing evidence by identifying their related adverse health outcomes, (4) facilitating healthcare providers and researchers in making informed decisions, (5) contributing to linking social health science to public health and health sciences.

## LITERATURE SURVEY METHODOLOGY

The current review followed a rigorous systematic method to ensure comprehensive and unbiased coverage of the literature by employing a rigorous review protocol and focusing on empirical evidence. A systematic review, thematic analysis, and tabular and figural analysis based on the PRISMA guideline by *Moher et al. (2009)* were adopted in this review study. The review aimed to address the following research questions: (1) What are the existing screening tools for assessing social frailty syndrome in older people living in Asia-Pacific region? (2) What are the main domains included in these social frailty screening tools? (3) What are the adverse health outcomes related to the social frailty domains in older people in Asia-Pacific region? PROSPERO was searched to ensure a similar systematic review study protocol was not registered. No prior studies focusing on the current topic of interest were identified. Thus, the current systematic search protocol was registered with PROSPERO (registration number: CR42021225980). PRISMA was used in the first and second steps of systematic review for data identification, screening, and eligibility, see Fig. 1. Thematic analysis (using theme, category, and code) and tabular and figural analysis were used for data synthesis and analysis of the final selected studies. Thematic, tabular, and figural analyses were conducted using Atlas.ti.9. Sarah Abdulkareem Salih and Siti Farra Zillah Abdullah performed the Search Strategy and any disagreements were resolved by the decision of a third reviewer (Tengku Aizan Hamid), using the Newcastle-Ottawa Scale (NOS).

### Search strategy

The first online search was conducted in January 2021 for studies over the last 20 years. The search was updated in February 2022 and 2023 to ensure the inclusion of recent studies conducted between January 2021 and January 2023. Two reviewers independently searched three electronic databases: PubMed, Scopus, and ScienceDirect; a manual search was conducted for the references of the complete full-text content using Google Scholar. The following search string and MeSH terms were used in the current systematic review: ("social frailty" OR "frailty social domain" OR "frailty social factor" OR "social vulnerability" OR "social vulnerabilities") AND ("frailty screening tools" OR "frailty assessment tools" OR instruments OR "assessment tools" OR "screening tools") AND ("risk factors" OR negative impacts OR "adverse outcomes" OR "adverse health outcomes") AND ("Aged" (MeSH Terms) OR "older adults" (MeSH Terms) OR elderly (MeSH Terms)) AND (Asia-Pacific (MeSH Terms) OR "Southeastern Asia communit*" (MeSH Terms)).

Truncation, Boolean operators, parentheses, wildcards, quotation marks, and MeSH terms or search terms related to the described keywords were applied whenever possible. The search string was used only for the title, abstract, and keywords. Figure 1 shows the PRISMA flow diagram for the steps of the systematic literature search.

### Inclusion and exclusion criteria

The study's inclusion and exclusion criteria were: (a) publication year and language of the study: including only studies published between 2002 and 2023 in the English language. Studies published before 2002 in a language other than English were excluded. According
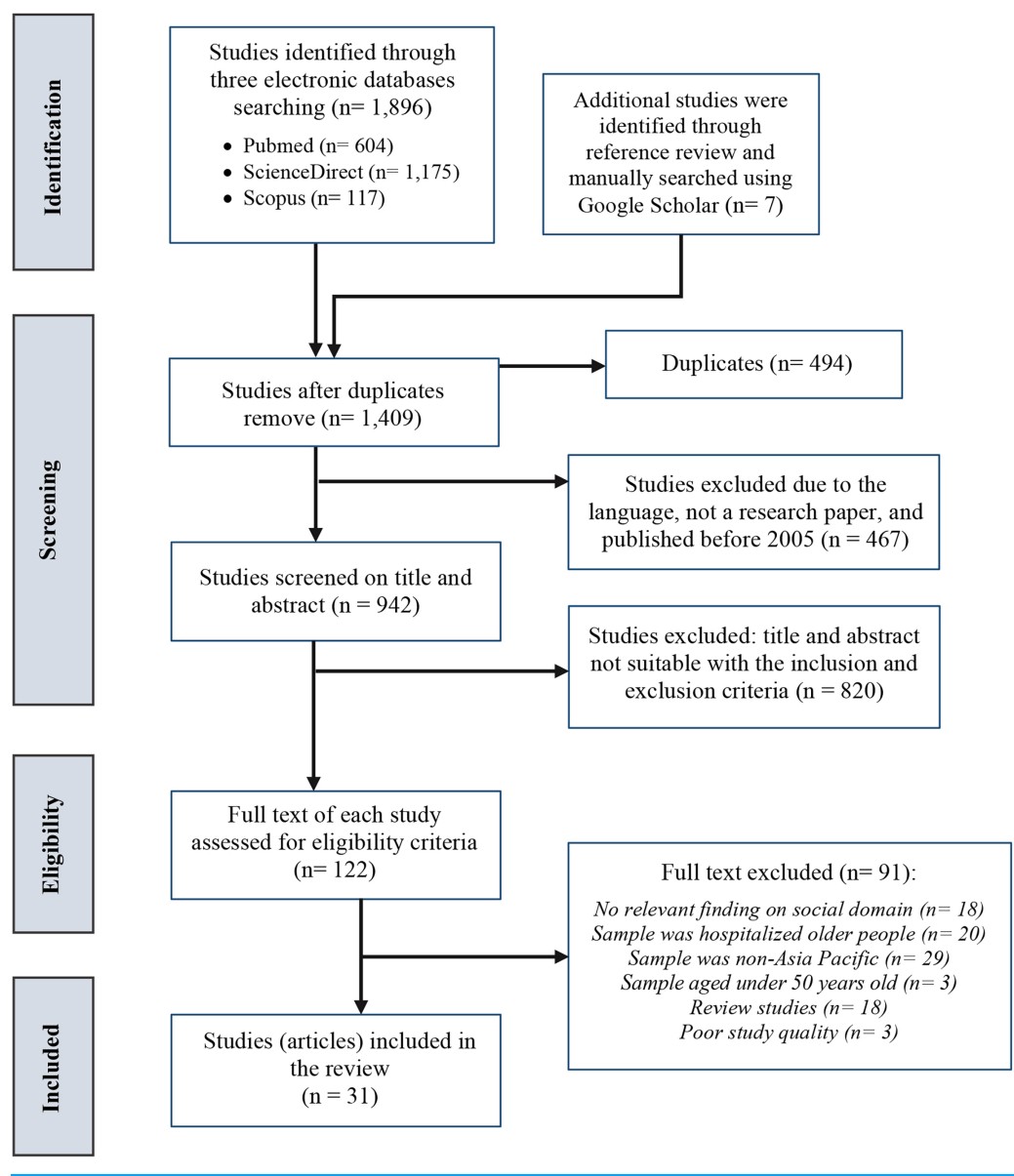

**Figure 1 PRISMA flowchart for the selected studies.**

to *Bessa, Ribeiro & Coelho (2018)*, social frailty is an emerging health-related topic, with most of the related literature evolving in the last 20 years. (b) Study type: including only empirical evidence published in peer-reviewed journals (*e.g.*, quantitative, qualitative, or mixed-methods). Review studies, books, reports, or other text materials not published in peer-reviewed indexed journals were excluded to ensure the quality of the selected studies. (c) Location: The studies' location must be in the Asia-Pacific region; this is based on the objective of the review. (d) Study scope: selected studies must include the assessment of the social domain of frailty. Studies that did not include any social domain of frailty were excluded to ensure that the review question was addressed. (e) Target group: The sample of the studies must be community-dwelling older adults aged 60 years or older. Studies from the Asia Pacific identified the frailty syndrome of older people aged 60 and over based on

retirement possibilities; people aged 60 and above are defined as senior citizens in many Asia Pacific countries (*Lian et al., 2020*; *Pek et al., 2020*).

## Study selection

In PRISMA's first and second steps (identification and screening), two reviewers (Author 2 and Author 3) independently reviewed the articles' titles and abstracts to include relevant articles. Studies written in English between 2002 and 2023 were included, and duplicates were excluded. In the title and abstract screening, studies were included if they described and had the social frailty determinants, assessment tools, and domains or if they included a combination of these variables. Then, all articles selected by either one reviewer or both of them were included in the next step.

In the second step of PRISMA (eligibility), the included articles from the previous step were retrieved in full text from the same reviewers using the eligibility criteria mentioned above. A manual search was conducted on the reference lists of the selected articles to avoid any missing articles in the search process during the previous step. Ultimately, the two independent reviewers engaged in a comprehensive discussion regarding the articles and reached a consensus on their inclusion. If an agreement could not be achieved, the ultimate determination was made by a third impartial reviewer (Author 1) using the Newcastle-Ottawa Scale (NOS). After selecting the included studies, a summary table was reported to present data from the studies, see Document S2. The included studies were then analyzed and synthesized by the two authors using Atlas.ti.9. All five authors had approved the steps of the search protocol.

## Data extraction and synthesis

The two reviewers extracted all data from the selected articles that could be related to social frailty, its domains, health outcomes, and assessment tools. To include a broad scope of factors related to social frailty, the data extracted is based on the definition and conceptual framework of social frailty. For example, each criterion included in the existing frailty index and showed an effect on social frailty was reviewed, such as socioeconomic status, loneliness, or isolation. *Gobbens et al.*'s *(2010)* and *Bunt et al.*'s *(2017)* frameworks were used to synthesize the criteria found and based on the factors highlighted in the conceptual framework of the current study. This conceptual background was used to have a comprehensive framework that integrates all domains and assessment tools associated with social frailty. Therefore, three themes to extract the data were (a) frailty assessment tools, (b) social domains of frailty, and (c) adverse outcomes related to social frailty and its domains. The three themes were identified based on the systematic review questions and objectives. For the first theme (frailty assessment tools), the data was synthesized and analyzed in sixteen categories describing each assessment tool's main definition and social frailty domains. The data was synthesized in six codes and four categories in the second theme (social domains of frailty). The third theme (adverse outcomes related to social frailty and its domains) included eleven codes (health outcome) and four categories. The four categories of the second and third themes included (1) social resources, (2) social

**Table 1 The social domain of frailty assessment tools.**

| Frailty assessment tool | Source | Freq. | Items No. | Category | Codes: social domain | Social domain item(s) | Scale | No. (%) of social frailty |
|---|---|---|---|---|---|---|---|---|
| Kaigo-Yobo Checklist (KYCL) | *Abe et al. (2019)* | 1 study | 15 | Social Behaviour Social Resources | Social activities: Social network: | 1) How often do you usually go out? <br> 2) Do you usually stay at home all day long? <br> 3) Do you have any hobbies? <br> 4) Do you have neighbors who you can talk closely with? <br> 5) Do you have friends, family, or relatives who you visit? | Dichotomous (2 points) | 5 (33.3%) |
| Social Frailty Questionnaire (SFQ) for Makizako et al. | *Makizako et al. (2015)* | 13 studies | 5 | Social Behaviour Social Needs Social Resources | Social activities: Social role: Social network: | 1) Do you go out less often now than you did a year ago? <br> 2) Do you rarely visit the homes of your friends? <br> 3) Do you feel that you are unhelpful to family/friends? <br> 4) Do you live alone? <br> 5) Do you not have the opportunity to speak to other daily? | Dichotomous (2 points): "yes" or "no" | 5 (100.0%) |
| Kihon Checklist (KCL) | *Watanabe et al. (2020)* | 1 study | 25 | Social Behaviour Social Needs | Social activities: Social support: | 1) Do you sometimes visit your friends? <br> 2) Do you go out at least once a week? <br> 3) Do you go out less frequently compared to last year? <br> 4) Do you turn to your family or friends for advice? | Dichotomous (2 points): "yes" or "no" | 4 (16.0%) |
| The Chinese version of the Lubben Social Network Scale (LSNS-6) | *Kuo et al. (2019)* | 1 study | 6 | Social Resources Social Needs | Social network: Social support: | 1) How many relatives do you see or hear from at least once a month? <br> 2) How many of your friends do you see or hear from at least once a month? <br> 3) How many relatives do you feel close to such an extent that you could call on them for help? <br> 4) How many relatives do you feel at ease with that you can talk to about private matters? <br> 5) How many friends do you feel at ease with that you can talk about private matters? <br> 6) How many friends do you feel close to such that you could call on them for help? | Ordinal (5 points): "0 = none", "1 = one", "2 = two", "3 (three/four)", "4 (five-eight)", and "5 (nine/more) | 6 (100.0%) |
| Social Frailty Questionnaire (SFQ) for Lian et al. | *Lian et al. (2020)* | 1 study | 2 | Social Resources Social Behaviour | Social network: Social activities: | 1) How often in the last month did you contact your friends? <br> 2) How often in the last month did you participate in a social activity? | Ordinal (5 points) | 2 (100.0%) |
| Social Frailty Screening Questionnaire (HALFT)[1] by Ma et al. | *Ma, Sun & Tang (2018)* | 1 study | 5 | Social Needs Social Behaviour Social Resources General Resources | Social role: Social activities: Social network: Financial difficulty: | 1) Are you able to help others within the past 12 months? <br> 2) Had you engaged in any social or leisure time activities in the previous 12 months? <br> 3) Do you have anyone to talk to? <br> 4) Are you living alone? <br> 5) Is your income was enough over the past 12 months? | Dichotomous (2 points): "yes" or "no" | 5 (100.0%) |

| Frailty assessment tool | Source | Freq. | Items No. | Category | Codes: social domain | Social domain item(s) | Scale | No. (%) of social frailty |
|---|---|---|---|---|---|---|---|---|
| Social Deficits/Social Frailty Questionnaire (SD-SF) by Lee et al. | *Lee et al. (2020)* | 1 study | 5 | General Resources Social Resources Social Needs Social Behaviour | Socioeconomic: Financial difficultly: Social network: Social support: Social activities: | 1. Education<br>2. Household income<br>3. Marital status<br>4. Family structure (Living alone)<br>5. The number of close relatives, friends, or neighbors<br>6. Contact relatives, friends, or neighbors<br>7. Emotional (listening to concerns or worries)<br>8. Instrumental (help with housework, cooking, *etc.*)<br>9. Care (support received from others)<br>10. Engagement in 8 types of activities | Dichotomous (2 points) | 5 (100.0%) |
| Social Vulnerability Index (SVI) by Armstrong et al. | *Armstrong et al. (2015)* | 1 study | 17 | Social Resources Social Needs Social Needs Social Behaviour General Resources | Social network: Social support: Social role: Social activities: Financial difficulty: | 1. When lonely, there is no one to talk to<br>2. See few many relatives once a month<br>3. Feel close to few people or relatives<br>4. No close friends<br>5. I rarely meet/talk with family or friends<br>6. Living alone<br>7. Present marital status<br>8. Cannot find somebody help w/ daily chore<br>9. Do not know somebody can turn to with personal issues<br>10. Do not trust at least one person's advice<br>11. Could not find someone to care for house<br>12. Do not have somebody to talk about important decision<br>13. Don't have people to help with things like shopping<br>14. People do not talk to you about important decisions<br>15. Do not participate in any groups<br>16. Do not do regular volunteer work<br>17. Low yearly household income | Dichotomous (2 points): "0 = absent" or "1 = present" | 17 (100.0%) |
| Social Frailty Questionnaire (SFQ) for Teo et al. | *Teo et al. (2019)* | 3 studies | 8 | Social Resources Social Needs Social Behaviour General Resources | Social network: Social support: Social activities: Socioeconomic: Financial difficulty: | 1) Do you live alone?<br>2) Infrequent visits or calls with others?<br>3) Do you have someone to confide in?<br>4) Receives a little help when required.<br>5) Infrequent social activities across various activity categories<br>6) No education?<br>7) Housing type<br>8) Are you limited by your financial resources to pay for needed medical service? | Dichotomous (2 points) Ordinal (5 points) Dichotomous (2 points) Nominal (3 points) | 7 (100.0%) |
| Social Frailty Questionnaire (SFQ) for Yamada and Arai | *Yamada & Arai (2018)* | 3 studies | 4 | Social Resources General Resources Social Behaviour | Social network: Financial difficulty: Social activities: | 1) Do you live alone?<br>2) How do you get along with your neighbors?<br>3) Are you satisfied with your economic condition?<br>4) Which social activities do you participate in? | Dichotomous (2 points) Ordinal (4 points) Nominal (4 points) | 4 (100.0%) |

*(Continued)*

| Frailty assessment tool | Source | Freq. | Items No. | Category | Codes: social domain | Social domain item(s) | Scale | No. (%) of social frailty |
|---|---|---|---|---|---|---|---|---|
| Tilburg Frailty Indicator (TFI) | *Song et al. (2020)* | 1 study | 25 | Social Resources Social Needs | Social network: Social support: | 1) Do you live alone?<br>2) Do you sometimes miss having people around you?<br>3) Do you receive enough support from other people? | Dichotomous (2 points): "yes" or "no" | 3 (20.0%) |
| Chinese-Comprehensive Frailty Assessment Instrument (CFAI) for Qiao et al. | *Qiao et al. (2018)* | 1 study | 21 | Social Needs | Social support: | 1) There are enough people whom I can rely on when I am in trouble.<br>2) I know many people whom I can totally trust<br>3) There are enough people with whom I feel a bond<br>4) Whom would you be able to appeal to? | Ordinal (5 points) Nominal (3 points) | 6 (26.1%) |
| Social Frailty Questionnaire for (SFQ) Pek et al. | *Pek et al. (2020)* | 1 study | 9 | Social Behaviour Social Needs Social Resources Social Needs General Resources | Social activities: Social role: Social network: Social support: Financial difficulty: | 1) Do you go out less frequently compared with last year?<br>2) Do you sometimes visit your friends?<br>3) Do you feel you are helpful to friends or family?<br>4) Do you live alone?<br>5) Do you talk with someone every day?<br>6) Do you eat with someone at least once a day?<br>7) Do you turn to family or friends for advice?<br>8) Do you have someone to confide in?<br>9) Are you limited by your financial resources? | Dichotomous (2 points) | 9 (100.0%) |
| Frailty Assessment Scale (FAS) | *Kim et al. (2021)* | 1 study | 14 | Social Needs Social Behaviour Social Resources | Social support: Social activities: Social network: | 1) Is there someone available to you whom you can count on to listen to you when you need to talk?<br>2) Do you participate in any of the following meetings or activities?<br>3) Do you have as much contact as you would like with someone you feel close to? | Dichotomous (2 points): "yes" or "no" | 3 (21.4%) |
| Chinese version of Comprehensive Geriatric Assessment-Frailty Index (CGA-FI) (*Ma et al., 2017*) | *Ma et al. (2017)* | 1 study | 14 | Social Needs | Social support: | Uses formal home supports? | Dichotomous (2 points) | 1 (7.1%) |
| Comprehensive Model of Frailty (CMF) | *Kwan, Lau & Cheung (2015)* | 1 study | 44 | Social Resources Social Behaviour Social Needs | Social networks: Social activities: Social support: | 1) Do you live alone?<br>2) Do you attend regular social activities (weekly)?<br>3) Do you have someone to confide in? | Dichotomous (2 points): "yes" or "no" | 3 (6.8%) |

**Note:**

[1] HALFT: an acronym for the five components "Help", "Participation", "Loneliness", "Financial" and "Talk".

needs, (3) social behaviors (or social activities), and (4) general resources, see Tables 1 and 2.

## Study quality assessment

The two reviewers (Author 2 & Author 3) independently carried out a quality assessment of the included studies using the Newcastle-Ottawa Scale (NOS) (*Wells et al., 2000*). Any disagreements regarding the included studies were resolved in an online meeting among the authors. The impartial reviewer (Author 1) did the final review using the

**Table 2 The social domain of frailty and their related adverse health outcomes.**

| Category | Subcategory: social domain | No of study | No of tools | Frailty assessment tools | Codes: significant health outcomes[1] |
|---|---|---|---|---|---|
| Social Resources | Social network | 29 | 13 | KYCL; SFQ for Makizako et al.; Chinese-LSNS-6; SFQ for Lian et al.; HALFT; SD-SF; SVI; SFQ for Teo et al.; SFQ for Yamada & Arai; TFI; CFAI; SFQ for Pek et al.; FAS | 1) Mortality 2) Physical frailty (mobility) 3) Cognitive frailty 4) Oral frailty 5) Mental illness (depression) 6) Dementia 7) Disability incidence 8) Chronic pain 9) Hearing loss 10) Nutritional disorders 11) BMI (or obesity) |
| Social Behaviour | Social activities | 28 | 12 | KYCL; SFQ for Makizako et al.; KCL; SFQ for Lian et al.; HALFT; SD-SF; SVI; SFQ for Teo et al.; SFQ for Yamada & Arai; SFQ for Pek et al.; CMF; FAS | 1) Mortality 2) Physical frailty (mobility) 3) Cognitive frailty 4) Oral frailty 5) Mental illness (depression) 6) Dementia 7) Disability incidence 8) Chronic pain 9) Hearing loss 10) Nutritional disorders 11) BMI (or obesity) |
| Social Needs | Social role | 16 | 4 | SFQ for Makizako et al.; HALFT; SVI; SFQ for Pek et al. | 1) Mortality 2) Physical frailty (mobility) 3) Cognitive frailty 4) Oral frailty 5) Mental illness (depression) 6) Dementia 7) Disability incidence 8) Chronic pain 9) Nutritional disorders |
| Social Needs | Social support | 13 | 11 | KCL; Chinese-LSNS-6; SD-SF; SVI; SFQ for Teo et al.; TFI; CFAI; SFQ for Pek et al.; CMF; FAS; CGA-FI | 1) Mortality 2) Physical frailty (mobility) 3) Mental illness (depression) 4) Disability incidence 5) Nutritional disorders 6) Hearing loss 7) BIM (or obesity) |
| General Resources | Financial difficulty | 10 | 6 | HALFT; SD-SF; SVI; SFQ for Teo et al.; SFQ for Yamada & Arai; SFQ for Pek et al. | 1) Mortality 2) Physical frailty (mobility) 3) Cognitive frailty 4) Mental illness (depression) 5) Disability incidence 6) Nutritional disorders 7) Dementia |
| General Resources | Socioeconomic | 4 | 2 | SD-SF; SFQ for Teo et al. | 1) Mortality 2) Physical frailty (mobility) 3) Cognitive frailty 4) Disability incidence |

**Note:**
[1] Each assessment tool predicted one or more than one adverse health outcome, for example, the Comprehensive Geriatric Assessment-Frailty Index (CGA-FI) measured the social support of social frailty and predicted three health outcomes: depression, mobility, and disability incidence; for more details, refer to Document S2 (*Makizako et al., 2015*; *Lian et al., 2020*; *Teo et al., 2019*; *Yamada & Arai, 2018*; *Pek et al., 2020*).

Newcastle-Ottawa Scale (NOS) to solve the disagreement and decide on the included study. The NOS is a risk-of-bias tool for nonrandomized studies, which assesses three domains, including study selection (four factors), study comparability (one factor), and exposure and outcomes (two factors). In general, studies with a low risk of bias (NOS score from 6 to 7 points) were included in the current review. Studies with a high risk of bias (NOS score from 4 to 5 points) and a very high risk of bias (NOS score from 0 to 3 points) were excluded (*Wells et al., 2000*), see Document S3.

## RESULTS

In the first step, a total of 1,896 text materials were initially identified through the search using the mentioned databases. Besides, through manual search, seven additional materials were included from the available reference lists of the final selected text materials. After eliminating duplicate studies and including journal articles published in English between 2002 and 2023, 942 studies remained; however, 820 were excluded based on title and abstract. The title or the abstract of these 820 excluded studies showed that the scope of the studies did not include the social domain of frailty ($n = 426$), the study location was not conducted in the Asia-Pacific region, including East Asia, Southeast Asia, and Oceania, ($n = 256$), or the study sample was not a community-dwelling older adults ($n = 138$). In the second step, a full text of 122 studies was read by the two reviewers, leading to the inclusion of 31 studies and the exclusion of 91 studies, as they included frailty in countries other than the Asia-Pacific countries ($n = 29$), involved samples of less than 60 years of age ($n = 3$), based on retirement age in many Asia Pacific countries, sample does not include community-dwelling older people ($n = 20$), the scope of the studies did not include the social domain of frailty ($n = 18$), were review studies ($n = 18$), and failed in quality appraisal (NOS total quality score was less than 6; $n = 3$). The rejected studies mostly could not provide information on outcomes, ascertainment of exposure, controlled confounding factors, or sample size. A PRISMA flowchart of the selection process is shown in Fig. 1, and PRISMA chart is shown in Document S1. The results were analyzed and synthesized based on three mentioned themes.

### Selected study characteristics

For the individual studies, 11 were cross-sectional studies (secondary data analysis), 13 were longitudinal studies (secondary data analysis), six were cross-sectional surveys (primary data analysis), and one was the experts' interview method (preliminary data analysis). These studies were conducted in five Asia-Pacific countries, including Japan ($n = 17$), China ($n = 6$), South Korea ($n = 4$), Singapore ($n = 2$), and Thailand ($n = 1$). Besides that, one of the selected studies was conducted in a general Asia-Pacific community of older adults without naming the country where the study was conducted. Of these 31 selected articles, 16 frailty assessment instruments with relevant social indicators were reviewed. Eight frailty instruments were published/designed in Chinese, five published in Japanese, two in Korean, and one in English. For more details of the individual study, Document S2 provided a detailed summary of the included study.

## Frailty assessment tools

This section describes the first theme regarding the assessment tools designed to measure social frailty syndrome in older adults in Asia-Pacific region. It reports sixteen frailty assessment tools ($n = 16$ tools) and six social domains of frailty (six codes), see Table 1. The most used tool ($n = 13$ studies) was the Social Frailty Questionnaire (SFQ) by *Makizako et al. (2015)*, a five-item self-reported questionnaire used to assess three social domains: social activities ($n = 2$), social role ($n = 1$), and social network ($n = 2$). Three studies mentioned the Social Frailty Questionnaire (SFQ) by *Teo et al. (2017)* that measured five social domains: social network ($n = 2$), social support ($n = 2$), social activity ($n = 1$), socioeconomic ($n = 2$), and financial difficulty ($n = 1$). Three studies also used the Social Frailty Questionnaire (SFQ) by *Yamada & Arai (2018)*, a four-question screening tool that consisted of three domains: social activity ($n = 1$), social network ($n = 2$), and financial difficulty ($n = 1$).

The following assessment tools were mentioned in one study each. The Social Frailty Questionnaire (SFQ) by *Lian et al. (2020)* assessed two domains (daily social activities and social network). *Pek et al. (2020)* also designed a nine-item Social Frailty Questionnaire (SFQ) that determined five domains of social frailty, including support ($n = 3$), social network ($n = 2$), social activities ($n = 2$), social role ($n = 1$), and financial difficulty ($n = 1$). *Armstrong et al. (2015)* designed the Social Vulnerability Index (SVI) that measures five social domains, including social network ($n = 7$), social support ($n = 6$), social activity ($n = 2$), social role ($n = 1$), and financial difficulty ($n = 1$). The Chinese version of the Lubben Social Network Scale (LSNS-6) by *Lubben et al. (2006)* was a six-item self-reported scale translated by *Kuo et al. (2019)*. It comprised two domains, namely social network ($n = 2$) and social support ($n = 4$). The Social Frailty Screening Questionnaire (HALFT) tool was designed by *Ma, Sun & Tang (2018)* and consisted of four social domains: social networks ($n = 2$), social role ($n = 1$), social activity ($n = 1$), and financial difficulty ($n = 1$). The Social Deficits/Social Frailty Questionnaire (SD-SF) was a ten-item scale designed by *Lee et al. (2020)* to measure five domains, including social networks ($n = 4$), social support ($n = 3$), socioeconomic ($n = 1$), financial difficulty ($n = 1$), and social activity ($n = 1$).

*Kwan, Lau & Cheung (2015)* developed the Comprehensive Model of Frailty (CMF), which consisted of a 44-item frailty scale measuring frailty syndrome among Chinese older adults. The social frailty domain included three social items: social networks ($n = 1$), social activities ($n = 1$), and social support ($n = 1$). The Comprehensive Geriatric Assessment-Frailty Index (CGA-FI) was a 14-item scale measuring five domains of frailty developed by *Ma et al. (2017)*; its social domain measured a dichotomous item related to social support ($n = 1$). *Qiao et al. (2018)* developed the Chinese-Comprehensive Frailty Assessment Instrument (CFAI), which is a 21-item self-report frailty tool. *Kim et al. (2021)* developed the Frailty Assessment Scale (FAS); this 14-item self-report frailty assessment tool assessed three domains of frailty, the social domain measured three items regarding social activities ($n = 1$), social support ($n = 1$), and social networks ($n = 1$). Meanwhile, *Abe et al. (2019)* designed the Kaigo-Yobo Checklist (KYCL), a 15-item dichotomous scale. Its social domain provided five questions measuring social activities ($n = 3$) and social

networks ($n$ = 2). The total social domain score ranged from 0 to 5, with a higher score indicating more social activity. The Kihon Checklist (KCL) was another dichotomous tool developed by the Japanese Ministry of Health, Labour, and Welfare reported by *Watanabe et al. (2020)*. The social domain provided four items only assessing social activities ($n$ = 3) and social support ($n$ = 1). The Tilburg Frailty Indicator (TFI) was a 25-item self-report questionnaire reported by *Song et al. (2020)*. Its social frailty domain consisted of a dichotomous scale for social networks ($n$ = 2) and social support ($n$ = 1).

## Social domains of frailty

Regarding the individual domains of social frailty, the current review yielded six potential domains (codes) for social frailty within the four categories, see Table 1. The social network was the most utilized domain in the reviewed scales ($n$ = 13 screening tools; $n$ = 29 study), followed by social activities ($n$ = 12 screening tools; $n$ = 28 study), social support ($n$ = 11 screening tools; $n$ = 13 analysis), financial difficulty ($n$ = 6 screening tools; $n$ = 10 studies), social role ($n$ = 4 screening tools; $n$ = 16 studies), and sociodemographic ($n$ = 2 screening tools; $n$ = 4 studies), such as education status and housing type. Therefore, social resources and social needs were the most frequent category in the reviewed tools ($n$ = 13 tools each), followed by social behaviors ($n$ = 12 tools) and general resources ($n$ = 6 tools).

The social network domain refers to the availability of physical, emotional, or virtual (*e.g.*, online) frequent contact with someone. It consisted of a wide range of questions about frequent contact with others, feeling close, or living with someone. Social activities refer to the regular (group) activities of older adults, and they include items related to the frequency of going out, doing a workout, or having a hobby (*Ma, Sun & Tang, 2018*; *Watanabe et al., 2020*).

Furthermore, social support represents confiding or turning to others for advice, help, care, and support (*Armstrong et al., 2015*; *Pek et al., 2020*). The social role included items related to feeling helpful to others (*Makizako et al., 2018b*; *Tsutsumimoto et al., 2018*). As for financial difficulty, the questions were about receiving enough income or feeling satisfied with the economic status (*Lee et al., 2020*; *Pek et al., 2020*), see Table 1. Four studies only showed that females have significantly higher levels of social frailty related to social activity, social support, and social network domains than males (*Abe et al., 2019*; *Kuo et al., 2019*; *Lee et al., 2020*; *Song et al., 2020*). However, the other studies didn't provide significant results regarding social frailty domains based on gender differences.

## Adverse outcomes related to social frailty and its domains

This section investigates the adverse health outcomes related to the severity of social frailty domains in community-dwelling Asia-Pacific older adults. The study revealed eleven adverse health outcomes ($n$ = 11 codes) associated with the social domains of frailty: mortality ($n$ = 6 domains), physical frailty or mobility issues ($n$ = 6 domains), disability incidence ($n$ = 6 domains), cognitive frailty ($n$ = 5 domains), mental illness or depression ($n$ = 5 domains), nutritional disorders ($n$ = 5 domains), dementia ($n$ = 4 domains), oral frailty ($n$ = 3 domains), hearing loss ($n$ = 3 domains), BMI or obesity issues ($n$ = 3 domains), and chronic pain ($n$ = 3 domains). Thus, all six mentioned domains of social

frailty predicted mortality, physical frailty, and disability incidence, see Table 2. Therefore, social resources, social needs, and social behavior predicted all eleven health outcomes, while general resources predicted only seven health outcomes, as shown in Table 2.

Social network and social activities domains predicted all eleven adverse health outcomes, see Table 2. Social role frailty predicted six health outcomes: physical, cognitive, and oral frailty, mental illness (depression), nutritional issues, dementia, disability incidence, chronic pain, and mortality. Social support frailty was associated with seven health outcomes, including physical frailty, mental frailty, disability incidence, nutritional disorders, hearing loss, BMI or obesity, and predicted mortality (*Armstrong et al., 2015*; *Watanabe et al., 2020*). The financial difficulty domain also predicted seven health outcomes: physical and cognitive frailty, mental illness or depression, malnutrition, dementia, disability incidence, and mortality (*Pek et al., 2020*). Moreover, the socioeconomic domains were associated with four health outcomes: physical frailty, cognitive frailty, disability incidence, and mortality (*Teo et al., 2017*); see Table 2. However, skeletal muscle mass had no significant association with these social domains (*Makizako et al., 2018a*; *Kim et al., 2021*; *Lian et al., 2020*). AD risk and EDS were confounding factors significantly associated with some social domains and non-significant associations with others (*Pek et al., 2020*; *Yamada & Arai, 2018*). Besides, each assessment tool predicted different health outcomes, see Document S2.

Two main methods were used to measure the reported health outcomes: clinical observation ($n$ = 8 health outcomes) and screening tools ($n$ = 4 health outcomes). Observation using medical diagnosis measured six health outcomes, including physical frailty, oral frailty, dementia, disability incidence, chronic pain, and hearing loss (*Abe et al., 2019*; *Armstrong et al., 2015*; *Hirase et al., 2019*; *Ma, Sun & Tang, 2018*; *Makizako et al., 2015*; *Tsutsumimoto et al., 2018*; *Nakakubo et al., 2019*; *Song et al., 2020*; *Watanabe et al., 2020*). Mortality was measured by observing the number of deaths, and obesity was measured using the observation of BMI (weight by the square of height). Screening tools using a questionnaire checklist measured four health outcomes: physical frailty (*e.g.*, using the fraility index), cognitive frailty (using the National Center for Geriatrics and Gerontology functional assessment tool (NCGG-FAT) or Mini-Mental State Examination (MMSE)), depression (using Geriatric Depression Scale (GDS) or Epidemiological Studies Depression (CESD)), and nutritional disorders (using Mini Nutritional Assessment (MNA) or Nutrition Screening Initiative (NSI)) (*Park et al., 2019*; *Pek et al., 2020*; *Teo et al., 2019*), see Table 3.

## DISCUSSION

As the authors know, this is one of the early systematic reviews conducted on a wide range of social domains of the Asia-Pacific frailty assessment tools and their adverse health outcomes. The findings of this review revealed a great diversity of social items arranged in six main domains included in the available Asia-Pacific screening tools for assessing frailty syndrome in community-dwelling older adults. These six domains were derived from the four aspects of social frailty definitions by *Bunt et al. (2017)* and *Gobbens et al. (2010)*, including social resources, social needs, social behavior or social activities, and general

**Table 3 Methods used to measure related adverse health outcomes.**

| Health Outcome | (%) of method used | Measurement method | Social domain predictors: category | Social domain predictors: subcategory |
|---|---|---|---|---|
| Mortality | 100% | Observation by measuring number of deaths | n = 4<br>Social Resources; Social Behaviour; Social Needs; General Resources | n = 6<br>Social network; Social activities; Social role; Social support; Financial difficulty; Socioeconomic |
| Physical frailty (mobility) | 80% | Observation: Medical diagnosis:<br>• Walking speed test (2.5-metre walk test)<br>• Grip strength test using Smedley Spring Dynamometers<br>• Time up and go test | n = 4<br>Social Resources; Social Behaviour; Social Needs; General Resources | n = 6<br>Social network; Social activities; Social role; Social support; Financial difficulty; Socioeconomic |
|  | 20% | Checklist & Index:<br>• IADL & ADL Checklist<br>• Frailty Index |  |  |
| Cognitive frailty | 100% | Clinical checklist:<br>• National Center for Geriatrics and Gerontology functional assessment tool (NCGG-FAT)<br>• Mini-Mental State Examination (MMSE) | n = 4<br>Social Resources; Social Behaviour; Social Needs; General Resources | n = 5<br>Social network; Social activities; Social role; Financial difficulty; Socioeconomic |
| Oral frailty | 100% | Observation:<br>Clinical examination | n = 3<br>Social Resources; Social Behaviour; Social Needs | n = 3<br>Social network; Social activities; Social role |
| Mental illness (depression) | 100% | Mental health checklist:<br>• Geriatric Depression Scale (GDS)<br>• Epidemiological Studies Depression (CESD) | n = 4<br>Social Resources; Social Behaviour; Social Needs; General Resources | n = 5<br>Social network; Social activities; Social role; Social support; Financial difficulty |
| Dementia | 100% | Observation:<br>Medical diagnosis | n = 4<br>Social Resources; Social Behaviour; Social Needs; General Resources | n = 4<br>Social network; Social activities; Social role; Financial difficulty |
| Disability incidence | 100% | Observation of:<br>• Medical diagnosis<br>• Long-Term Care System (LTCS) | n = 4<br>Social Resources; Social Behaviour; Social Needs; General Resources | n = 6<br>Social network; Social activities; Social role; Social support; Financial difficulty; Socioeconomic |
| Chronic pain | 100% | Observation:<br>Medical diagnosis | n = 3<br>Social Resources; Social Behaviour; Social Needs | n = 3<br>Social network; Social activities; Social role |
| Hearing loss | 100% | Observation of:<br>• Medical diagnosis<br>• Pure-tone audiometry using various frequencies of Hz | n = 3<br>Social Resources; Social Behaviour; Social Needs | n = 3<br>Social network; Social activities; Social support |

| Table 3 (continued) | | | | |
|---|---|---|---|---|
| Health Outcome | (%) of method used | Measurement method | Social domain predictors: category | Social domain predictors: subcategory |
| Nutritional disorders | 100% | Checklist:<br>• Mini Nutritional Assessment (MNA)<br>• Nutrition Screening Initiative (NSI) | *n* = 4<br>Social Resources; Social Behaviour; Social Needs; General Resources | *n* = 5<br>Social network; Social activities; Social role; Social support; Financial difficulty |
| BMI (or obesity) | 100% | Observation of:<br>Measuring BMI (weight by the square of height) | *n* = 3<br>Social Resources; Social Behaviour; Social Needs | *n* = 3<br>Social network; Social activities; Social support |

resources, see Fig. 2. The most common social variables reviewed in Asia-Pacific frailty tools are social networks, social activities, social support, financial difficulties, and social roles. Therefore, social resources and social needs were the most frequent categories in the reviewed tools, followed by social behaviors and general resources.

Besides, the socioeconomic dimensions were observed in some screening tools established more recently (*Pek et al., 2020*; *Teo et al., 2019*). It was noted that a significant relationship existed between social frailty and sociodemographic characteristics, including education, income, and household type (*Abe et al., 2019*; *Kuo et al., 2019*). Thus, sociodemographic variables could be integrated as one of the main components of social frailty screening tools (*Bessa, Ribeiro & Coelho, 2018*). Studies from China conducted by *Ma, Sun & Tang (2018)* pointed to a connection between social frailty and socioeconomic factors, as living alone, house ownership, and education level are increasingly higher as frailty rises. Similarly, a European study by *Silan, Caperna & Boccuzzo (2019)* also confirmed the possibility of socioeconomic as a key domain of older people's social frailty. Therefore, the current study considered socioeconomic as a domain of social frailty syndrome, especially in Asia-Pacific settings. However, social isolation was identified to have little expressiveness in the instruments of frailty evaluation considered in this study. Therefore, they were included within the social network variable. These findings contribute to the result reported by *Bunt et al. (2017)* that the inclusion of not only the potential lack of social resources and social needs but also the potential absence of social behaviors or social activities and general resources of self-management abilities should be considered as integral components of the concept of social frailty and its domains. However, *Bunt et al. (2017)* focused on the four social domains of frailty from a general perspective. Furthermore, in only four of the selected studies, it was observed that female exhibit notably elevated levels of social frailty in areas such as social activity, social support, and social network domains compared to male (*Abe et al., 2019*; *Kuo et al., 2019*; *Lee et al., 2020*; *Song et al., 2020*). On the contrary, the remaining studies did not yield significant findings regarding gender differences in various social frailty domains. These different results may be due to the type of screening tools and questions used to measure social frailty syndrome.

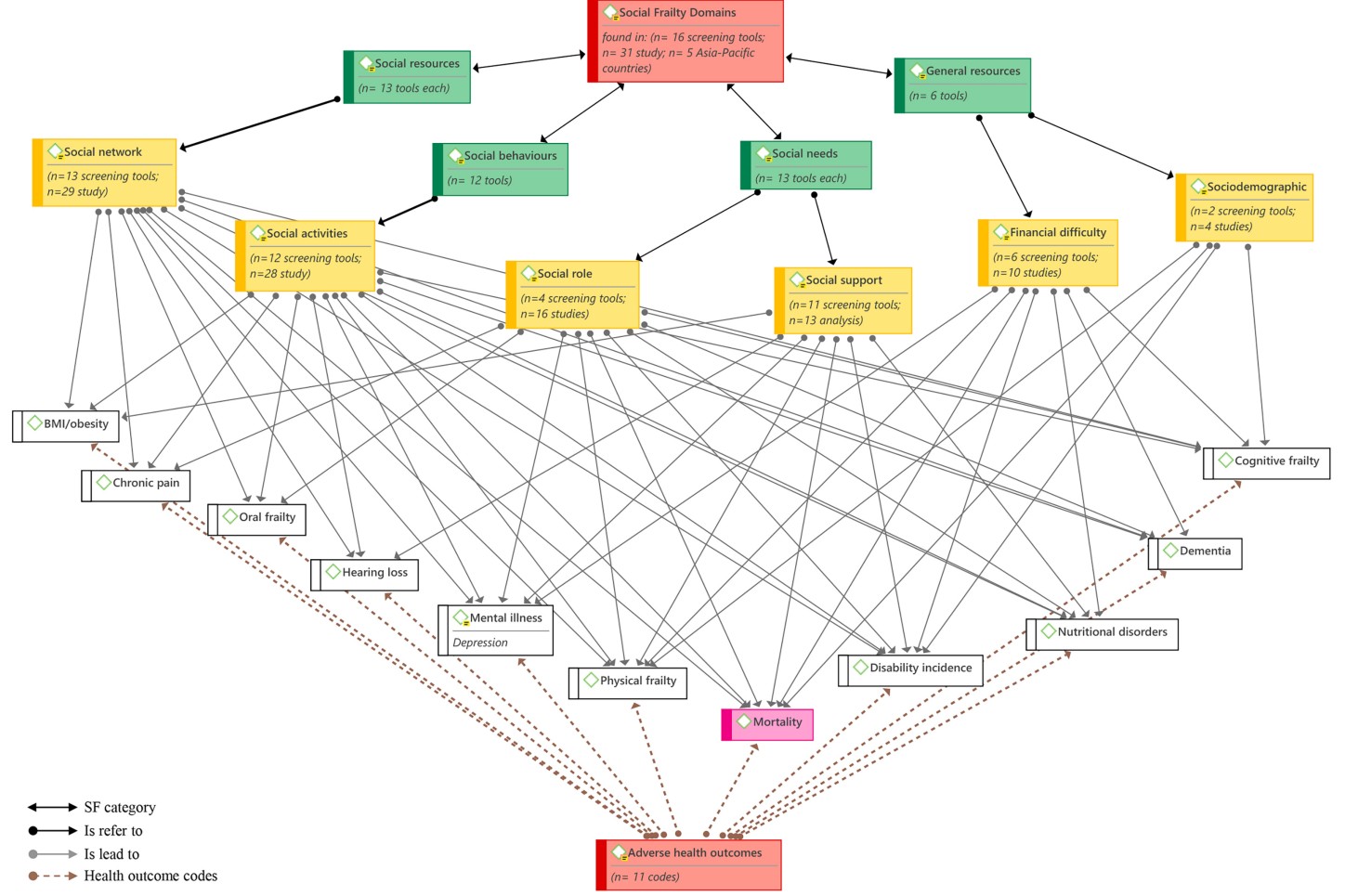

**Figure 2** Model of characterization of social frailty domains and related adverse health outcomes using Atlas.ti.9.

Regarding the frailty screening tools, the study reviewed sixteen frailty assessment tools with six social domains of frailty that assess social frailty syndrome in older people living in Asia-Pacific region. Generally, the social domain of these assessment instruments showed heterogeneous characteristics regarding the number of items, which varied between two and 17, see Table 1. The most frequently utilized scale was SFQ by *Makizako et al. (2015)*. This was reported in 13 studies, followed by SFQ by *Teo et al. (2017)* and SFQ for *Yamada & Arai (2018)* in three studies, while the other scales were observed in one study each. In addition, several studies reviewed used the SFQ for *Makizako et al. (2015)* where it demonstrated high efficiency in predicting adverse outcomes and mortality in community-living older adults (*Bae et al., 2018*; *Hironaka et al., 2020*; *Tsutsumimoto et al., 2017*). However, *Armstrong et al. (2015)* showed that the SVI was the only assessment tool capable and likely to cover a broader range of social domains and items. Nine scales, including SFQs (*n* = 5 Social Frailty Questionnaire for *Lian et al., 2020*; *Pek et al., 2020*; *Yamada & Arai, 2018*; *Teo et al., 2017*; *Makizako et al., 2015*), Chinese LSNS-6, HALFT,

FD-SF, and SVI, were 100% designed to measure the social frailty syndrome in older people, see Table 1.

It is also noted that four Asia-Pacific assessment scales contained a significant proportion of social items, including KYCL (33.3%), Chinese-CFAI (26.1%), FAS (21.4%), and TFI (20.0%). Consequently, limited social items were reviewed in physical and psychological frailty scales, such as KCL (16.0%), CGA-FI (7.1%), and CMF (6.8%). Two scales, including Chinese LSNS-6 (*Kuo et al., 2019*) and Japanese SVI (*Armstrong et al., 2015*), were developed from the existing social assessment tools; two other scales were also adapted from the current multidimensional frailty scales, including KYCL (*Abe et al., 2019*), and KCL (*Watanabe et al., 2020*). Besides that, four multidimensional frailty screening instruments were developed from the concept of the comprehensive frailty assessment instrument (*Bunt et al., 2017*), which included the Chinese version of CMF (*Kwan, Lau & Cheung, 2015*), CGA-FI (*Ma, Sun & Tang, 2018*), CFAI (*Qiao et al., 2018*) and FAS (*Kim et al., 2021*). The diversity in the Asia-Pacific social frailty screening tools represented a substantial percentage and weight to the social parameter by offering freedom of choice for the items. However, *Faller et al. (2019)* mentioned that the diversity of social frailty screening tools reflects the ambiguity in the conceptual definition and the operationalization of social frailty and can contribute to problems with evidence about the possible outcomes and syndrome prevalence. Therefore, the current study focused systematically on social frailty tools with specific periods and settings to contribute to an accurate definition of social frailty and its health outcomes. All the reviewed frailty instruments with their social domains provided clear evidence to enhance the precision of this definition. However, 25.80% of the studies reviewed did not provide precise data on the validity and reliability of the Asia-Pacific versions of social frailty screening tools (*Lian et al., 2020*; *Ma, Sun & Tang, 2018*; *Song et al., 2020*; *Teo et al., 2019*; *Yamada & Arai, 2018*). Based on *Kim et al. (2021)*, frailty assessment instruments and their characteristics must have a certain level of reliability and validity to be used.

The use of broader components was one of the relevant techniques in the context of frailty and social frailty tools to integrate frailty dimensions into the related healthcare tools (*Kwan, Lau & Cheung, 2015*). The main variables in the social frailty screening tools, including social networks, social activities, and social support, play a vital role in the general health of older adults by predicting a wide variety of adverse health outcomes such as physical and cognitive frailty, oral frailty, disability incidence, depression, dementia, nutritional disorders, hearing loss, chronic pain, and even mortality (*Abe et al., 2019*; *Hirase et al., 2019*; *Ma, Sun & Tang, 2018*; *Makizako et al., 2015*; *Tsutsumimoto et al., 2018*; *Nakakubo et al., 2019*; *Song et al., 2020*; *Watanabe et al., 2020*). Although there were different social domains with various items, they were often able to predict the mentioned adverse health outcomes (*Kim et al., 2021*; *Makizako et al., 2015*; *Nagai et al., 2020*; *Park et al., 2019*; *Pek et al., 2020*; *Teo et al., 2017*; *Yamada & Arai, 2018*). All of the social domains reviewed in the current study could predict mortality, physical difficulties, and disability incidence, referring to the critical role of social variables in older adults' frailty. Similarly, a study conducted by the National Health and Nutrition Examination Survey (NHANES) in the USA by *Zhang et al. (2018)* confirmed that frail people had higher

mortality and hospitalization rates than non-frail ones. However, this study focused on the gender-associated factors for frailty. The social domains had no significant association with skeletal muscle mass (*Makizako et al., 2018a*; *Kim et al., 2021*; *Lian et al., 2020*). However, *Kim et al. (2021)* and *Lian et al. (2020)* proved that skeletal muscle mass and muscle strength (grip strength) are key indicators of physical frailty. Therefore, muscle mass and muscle strength can be considered a variable for overall frailty; thus, enhancing body muscle mass and muscle strength is critical for reducing older people's physical frailty. AD risk and EDS were confounding factors that had a significant association with a few social domains and a non-significant association with others (*Pek et al., 2020*; *Yamada & Arai, 2018*); therefore, they were not included as a health outcome. The socioeconomic characteristics predicted fewer adverse outcomes (*Teo et al., 2019*). Previous evidence also confirmed the existence of a correlation between certain social aspects of frailty and oral problems (*Hironaka et al., 2020*) and BMI or obesity (*Song et al., 2020*; *Watanabe et al., 2020*).

Unlike social frailty, two methods were used to measure adverse health outcomes: clinical observation and screening tools. Clinical observation was the most used approach to measure seven health outcomes: mortality, physical frailty, oral frailty, dementia, disability incidence, chronic pain, hearing loss, and obesity (*Abe et al., 2019*; *Armstrong et al., 2015*; *Hirase et al., 2019*; *Ma, Sun & Tang, 2018*; *Makizako et al., 2015*; *Tsutsumimoto et al., 2018*; *Nakakubo et al., 2019*; *Song et al., 2020*; *Watanabe et al., 2020*). Followed by screening tools and checklist methods to check physical frailty (mobility), cognitive frailty, depression, and nutritional disorders (*Park et al., 2019*; *Pek et al., 2020*; *Teo et al., 2019*).

Accordingly, the current systematic review proposed a model for characterizing social frailty domains and related adverse outcomes based on a holistic overview of 16 screening tools for frailty syndrome from Asia-Pacific settings, see Fig. 2. It contributes to a comprehensive multi-dimensional definition of social frailty by providing a broad overview of the screening tools assessing social frailty in older people in the Asia-Pacific and by characterizing the related social domains and health outcomes. It explains social frailty as a lack of one or more aspects of social resources, social needs, social activities, and general needs, such as financial resources and socioeconomic difficulties, in older age, leading to several physical, mental, and cognitional health disorders and even mortality.

## Theoretical and practical implication

The current systematic review, based on the existing evidence and theories, such as the Social Production Function theory and social frailty concept by *Bunt et al. (2017)*, fills the gap in the literature by considering a wide range of social domains in the social frailty model and determining the related adverse health outcomes. The current systematic review contributes to the conceptual definition of social frailty and the interpretation and convergence of the concept of multidimensional social frailty. It also provides a better understanding of the role of each social domain in the conception of social frailty. Although the social variables of frailty, such as social activities and social resources, have been reviewed in the older people's frailty models (*Bessa, Ribeiro & Coelho, 2018*; *Bunt et al., 2017*), its related health outcomes are still unexplored, especially in the context of

Asia-Pacific settings. Thus, the current review contributed to the concept of social frailty, its domains, and its health outcomes in Asia-Pacific region. Therefore, the significance of this review lies in linking social health science to public health and health sciences.

The current study also proved that each domain of the six reviewed domains of social frailty predicts adverse health outcomes, including mortality. It enhanced our understanding of the expected adverse health outcomes resulting from the frailty of each social domain. This facilitates professionals and researchers in making informed decisions regarding selecting instruments tailored to each study's specific circumstances and scope. Understanding the social domains related to certain social frailty themes contributes to guiding healthcare professionals to develop appropriate interventions to mitigate or postpone the onset of specific older people's health outcomes and frailty. For example, proper intervention, such as enhancing social support combined with social activity and social role, can contribute to reducing nutrition disorders, depression, and disability, therefore decreasing frailty in older people. Social-based interventions, such as participation in social activities, may be beneficial for reducing social frailty among older people, especially in the Asia Pacific region. Therefore, professionals and researchers in health and social sciences are encouraged to consider the six mentioned domains to understand better the frailty syndrome and related health consequences. Furthermore, the current review provided a theoretical background for professionals and researchers to understand social frailty syndrome in older people.

## Limitations and future research directions

The limitations found in this review were the ambiguity between the adverse outcomes and the social domains, where some studies using the same frailty scale found a significant result about some of the possible outcomes, while others did not. One additional limitation identified in the study pertained to the potential confusion arising from the inconsistent nomenclature employed for specific scales across various research investigations. Moreover, many of the reviewed screening scales from the Asia-Pacific setting lacked or did not disclose clear validity and reliability checks. However, the authors used methodological rigor and only included quality studies by applying the quality assessment framework of *Wells et al. (2000)*. Finally, all the studies reviewed were written in English and published within the last 20 years. Therefore, authors excluded from the review investigations that may have been reported in other (Asian) languages and published before 2002 were excluded.

The current study is limited to social frailty assessed through questionnaires; this aligns with the study's objective to characterize the social frailty domains and their health outcomes by overviewing the frailty screening tools. However, experimental approaches, such as clinical practice and physical examinations, can contribute to a more objective frailty assessment. Therefore, future research is encouraged to conduct more objective methods for evaluating older people's frailty.

Based on these limitations, future research is recommended to develop a comprehensive empirical study to test the conceptual model of social frailty and its domains. A new comprehensive social assessment tool is required to contribute to the operationalization

framework and a unified conceptual definition of social frailty. Future interventions that target individuals or groups of socially frail older adults should also address the several domains of social frailty mentioned in this study. Besides, future studies must apply a certain level of reliability and validity to the assessment tools.

## CONCLUSION

The reviewed studies provide evidence for the conceptual definition of social frailty by examining a wide range of social frailty domains and their health outcomes among older adults in the Asia-Pacific region. It revealed six possible domains of social frailty: social networks, social activities, social support, financial difficulties, social roles, and socioeconomic, reviewed in sixteen frailty assessment tools. It also revealed that different social domains of social frailty could predict various adverse health outcomes, such as physical and cognitive frailty, oral frailty, depression, dementia, nutritional status, hearing loss, and even mortality. However, the health outcomes were non-homogeneous in all domains, where social network, social activity, and social support predicted the majority of adverse health outcomes compared to social role and socioeconomic variables. Yet, all of the six dimensions were able to predict mortality. These various components collectively contribute to satisfying social demands and better health. Therefore, social frailty can be conceptualized as a spectrum wherein individuals are susceptible to the potential loss or actual loss of social or general resources, activities, or capabilities crucial for meeting one or more fundamental social requirements throughout their lifespan. Understanding older people's social frailty contributes to predicting better health outcomes and quality of life. Identifying the adverse health outcomes related to certain domains of social frailty will help professionals, caregivers, and older people to apply the appropriate intervention.

### Funding
This study was part of the Transforming Cognitive Frailty to Later-Life Self-sufficiency (AGELESS) study funded by the Ministry of Higher Education Malaysia Long-Term Research Grant Scheme (LRGS/1/2019/UM/01/1/2), which has evolved from the LRGS TUA Study. The study was also supported by the Research Management Centre (RMC), Universiti Putra Malaysia (UPM). The funders had no role in study design, data collection and analysis, decision to publish, or preparation of the manuscript.

### Grant Disclosures
The following grant information was disclosed by the authors:
Ministry of Higher Education Malaysia Long-Term Research Grant Scheme: LRGS/1/2019/UM/01/1/2.
Research Management Centre (RMC), University Putra Malyasia (UPM).

### Competing Interests
The authors declare that they have no competing interests.

## Author Contributions

- Tengku Aizan Hamid conceived and designed the experiments, performed the experiments, analyzed the data, prepared figures and/or tables, authored or reviewed drafts of the article, and approved the final draft.
- Sarah Abdulkareem Salih conceived and designed the experiments, performed the experiments, analyzed the data, prepared figures and/or tables, and approved the final draft.
- Siti Farra Zillah Abdullah conceived and designed the experiments, performed the experiments, analyzed the data, authored or reviewed drafts of the article, and approved the final draft.
- Rahimah Ibrahim analyzed the data, authored or reviewed drafts of the article, and approved the final draft.
- Aidalina Mahmud analyzed the data, prepared figures and/or tables, and approved the final draft.

## Data Availability

The raw measurements are available in the Supplemental Files.

## Supplemental Information

Supplemental information for this article can be found online at http://dx.doi.org/10.7717/peerj.17058#supplemental-information.

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
