# Peer review of "Characterization of social frailty domains and related adverse health outcomes in the Asia-Pacific: a systematic literature review"

_PeerJ, doi:10.7717/peerj.17058_

## Round 0.1 · original submission · Minor Revisions

Congratulations. The reviewers found merit in the study and provided some important and relevant comments that should be considered. Please, we invite the authors to address the points raised during the review process.

Reviewer 1 ·

Basic reporting

None of the references were published after 2020. It is recommended to update the references.

Experimental design

Figure 2 is not clear enough. It is recommended to provide high-resolution pictures.
In Figure 2, the text is covered by lines. It is recommended to use different line types to express the meaning of the text.

Validity of the findings

Thirty-one studies were included in this paper based on the screening criteria, six social domains related to social frailty in older adults were analyzed, and all of these social domains were associated with adverse health outcomes.
This study provides all relevant basic data which are reliable, statistically sound, and the results and conclusions are clearly presented.

Reviewer 2 ·

Basic reporting

Dear Authors, please see some suggestions to improve the manuscript.

1. Please try to improve the abstract in methods.
Also try to add specific conclusions of your study. What you can contribute to a better aging process?

2. Literature are well presented but it can be improved

3. Please try to promote a better figure 3. It very difficult to analyze it.

Literature references and information are good. Just need a litle add information about the population, Asia-Pacific. If all the information and introduction is about the Asia-Pacific population, please consider to add it in the title of the manuscript.

Data and results are appropriate.

Discussion and conclusion can be improved.

Experimental design

The manuscript and research question are rigorous and the methods are well described.
Some points can be improved but the main characterization of experimental design are well presented.

Validity of the findings

Statistical point and results are well described but in results it seems that some information can be added.
Conclusions also can be improved. What people can change in their behaviour and what are the main considerations to politic strategy to people aged with more quality of life.

Additional comments

1. Please in the abstract add the main Asia-Paciûc communities. What is? some information can be added?
2. in the abstract is possible to add some information about the domains between female and male? Can also be added the intervale of ages of older people?
3. Also in the abstract try to change this keyword: physical defect,

In the introduction, when you add information about some studies, it is according to older people in Asia-Paciûc communities? It should be. Or try to add it if is possible. If you can make the manuscript the goal of the description of the characterization of social frailty domains and related adverse health outcomes in specific populations it could be great.

In the introduction, there are several information about social and community activities that can lead to social frailty. But what are the activities? Try to be more specific. there are a lot of information regarding frailty status and social conditions, but we talking about what?

Some of the information in line 127 can be added in the introduction.

In the results, clarify why 820 studies were excluded based on title and abstract.
Also, what it mean "did not contain relevant findings in the social domain of frailty (n=18)?
If is possible try to clarify the description of the results in all subtitles.
Also, why do you consider 60 and over to the review?
Do you also think that can be differences between females and male?

In discussion, in line 442 "the social domains had no significant association with skeletal muscle mas". What do you think about this? older people must perform physical exercise why? ...

In pratical aplications, line 480, you said "This also contributes to guiding them to develop therapies designed to mitigate or postpone the onset of social frailty." can you be more specific?
Also, it is possible to add some pratical strategies to decrease frailty?

·

Basic reporting

Research on frailty and adverse health outcomes, including mortality, physical, and mental disabilities, has been a topic of considerable interest over the last years. However, although several reports have highlighted the importance of this interdependent phenomenon, more information is required for the design of effective interventions and long-term implications for developing targeted healthcare strategies. English language used throughout is clear. The figures are relevant and clear.
This reviewer is quite satisfied with the significance of this study, since it contributes to understanding the health impact of social frailty. Potential supportive interventions to minimize social frailty side effects emerge from this study. Although the work raises some concerns which need to be further addressed.

The introduction contextualizes the problem, and the references seem relevant and appropriate
Live 48-50: “Contemporary research on frailty in older adults assumes a complex interaction between various dimensions, such as physical, cognitive, psychological, environmental, and social domains (Ma et al., 2016)” Please provide a revised version of the sentence. Describing the research as contemporary and referencing a source from 2016 does not seem appropriate.
Line 53-55- “Recent studies mentioned that social frailty could describe a lack of frequent participation in social events, networks, contact, and insufficient support, leading to serious health outcomes” Please provide a revised version of the sentence. Similarly, to the previous comment, referencing recent studies dating back to 2019 does not seem appropriate.

The conceptual background of the study refers to the theoretical framework and foundational concepts that underpin the research problem and helps establish the theoretical basis for the study.

Experimental design

The study's objective is clearly defined and relevant.
The need and relevance for undertaking a systematic review is clear by outlining gaps in existing knowledge.

Line 131. “ tool. (3)” This reviewer believes that period before (3) is a typo
Line 132-134 “ This contributed to facilitating healthcare providers and researchers in making informed decisions, (4) contributing to linking social health science to public health and health sciences.” Please improve the language since the current phrasing makes comprehension difficult.

The systematic approach and methods used to review, analyze, and synthesize existing literature on a specific topic or research question are well presented, including: Scope definition, search strategy and criteria and data extraction, synthesis and analysis.

Validity of the findings

Generally, the results and conclusions drawn from the study accurately and appropriately represent what the authors claim t characterized: social frailty domains and related adverse health outcomes. However, the work raises some concerns which need to be further addressed.

Results
Line 245-247 “exclusion of 91 studies, as they included (…)involved samples of hospitalized or patient older people (n= 20). Please clarify since this information is not on the section “Inclusion and exclusion criteria”

Frailty assessment tools
The instruments used to asses social frailty were validated to the respective sample application in the studies that were selected? In my opinion, this is an important issue.
From my understanding, in the selected studies, frailty was assessed through a questionnaire. This may be viewed as a limitation. Could more objective measures for evaluating frailty be considered? Discussing the methods used could make the article more interesting, as well as addressing the evaluation of health outcomes.

Adverse outcomes related to social frailty and its domains
What about health outcomes assessment? What tools were used to asses those outcomes in the different selected studies? Were the health outcomes assessed using questionnaires or more objective measures? In my opinion this should be clarified and discussed.

The tables are clear and provide relevant information for the understanding of the study. The conclusions are clearly articulated, directly tied to the original research question, and confined to supporting the obtained results.

In my ponion, "Theoretical and practical implications" and "Practical implications" should be merged.

---

## Round 0.2 · accepted · Accept

The authors have addressed all the reviewers' comments. One of the reviewers was not able to assess the revised version of the manuscript, but the comments provided were addressed. I am happy with the current version of the manuscript. Congratulations.

Reviewer 2 ·

Basic reporting

The authors revised the manuscript and it can be published.
The introduction and discussion were improved.The pictures were also improved.

Experimental design

The investigation in this population are very important and the experimental process are now well described and presented.

Validity of the findings

Conclusions were improved and also the main goal of the manuscript.

·

Basic reporting

The language throughout the text has been clarified, particularly at the points emphasized by the reviewers. The references have been updated. The figures and tables have been improved according to the suggestions.

Experimental design

The experimental design was clarified according to the reviewers' suggestions.

Validity of the findings

The results and particularly the discussion were improved according to the reviewers' comments.

Additional comments

no comment